# Overcoming Metabolic Constraints in the MEP-Pathway Enrich *Salvia sclarea* Hairy Roots in Therapeutic Abietane Diterpenes

**Mariaevelina Alfieri** †, **Alfredo Ambrosone**, **Mariacarmela Vaccaro, Nunziatina De Tommasi and Antonietta Leone** *

Department of Pharmacy, University of Salerno, 84084 Fisciano, Italy; m.alfieri@santobonopausilipon.it (M.A.); aambrosone@unisa.it (A.A.); mvaccaro@unisa.it (M.V.); detommasi@unisa.it (N.D.T.)
* Correspondence: aleone@unisa.it
† Current address: Clinical Pathology, Pausilipon Hospital, A.O.R.N Santobono-Pausilipon, 80123 Naples, Italy.

**Abstract:** Abietane diterpenoids (e.g., carnosic acid, aethiopinone, 1-oxoaethiopinone, salvipisone, and ferruginol) synthesized in the roots of several *Salvia* species have proved to have promising biological activities, but their use on a large scale is limited by the very low content extracted from in vivo roots. In this review, we summarized our efforts and the achieved results aimed at optimizing the synthesis of these diterpenes in *Salvia sclarea* hairy roots by either elicitation or by modifying the expression of genes encoding enzymes of the MEP-pathway, the biosynthetic route from which they derive. Stable *S. sclarea* hairy roots (HRs) were treated with methyl jasmonate or coronatine, or genetically engineered, by tuning the expression of genes controlling enzymatic rate-limiting steps (*DXS*, *DXR*, *GGPPS*, *CPPS* alone or in combination), by silencing of the *Ent-CPPS* gene, encoding an enzyme acting at gibberellin lateral competitive route or by coordinate up-regulation of biosynthetic genes mediated by transcription factors (WRKY and MYC2). Altogether, these different approaches successfully increased the amount of abietane diterpenes in *S. sclarea* HRs from to 2 to 30 times over the content found in the control HR line.

**Keywords:** *Salvia sclarea*; hairy roots; bioactive diterpenes; elicitation; metabolic engineering





## 1. Introduction

Medicinal plants are largely used as a source of natural remedies for preventing or combating chronic diseases. In addition, the identification of bioactive plant-derived molecules and the uncovering of their molecular targets provide an enormous opportunity for new drug development. The majority of the bioactive plant compounds are secondary metabolites, with low molecular weights (<1000 Da), which are synthesized in plants to promote their adaptation to both abiotic and biotic stresses, protection against herbivores and phytopathogens, and as attraction for pollinators and the dispersion of seeds mediated by animals. It is widely reported that the content of most of the bioactive secondary metabolites in wild or cultivated medicinal plants is generally very low (less than 1% dry weight) and variable, dependent on the physiological and developmental stages [1].

In addition, the complex chemical structures of medicinal plants make total chemical syntheses highly unrealistic. Although pharmaceutical companies have optimized the extraction of bioactive compounds or of their precursors from plants and developed semisynthetic processes, valuable plant-derived drugs, currently used in the clinic, usually have high costs of production. Plant biotechnology offers a sustainable method for the bioproduction of plant secondary metabolites using plant in vitro systems. However, there are still many challenges to overcome to enhance the production of these metabolites from plant in vitro systems and establish a sustainable large-scale biotechnological process [2].

Plant terpenoids constitute one of the most functionally and structurally diverse group of plant secondary metabolites thus far described, consisting of more than 80,000

different compounds, with diverse biological functions in the plant kingdom and for human health [3]. Many studies have been focused on elucidating the isoprenoid pathways, identifying the enzymatic steps, their genetic control, and uncovering potential enzymatic bottlenecks, with the final aim being metabolic bioengineering or refactoring a plant's natural product biosynthetic pathways in microorganisms [4].

Most plant isoprenoids, including also diterpenes, derive from two common $C_5$ precursors, IPP and its isomer DMAPP, through two distinct pathways: the well-studied mevalonate (MVA) pathway, which predominates in cytosol, and the more recently unveiled deoxyxylulose 5-phosphate/2-C-methyl-D-erythritol 4-phosphate (DXP/MEP) mevalonate-independent pathway, localized in the plastids (Figure 1). Briefly, the condensation of pyruvate and glyceraldehyde 3-phosphate (G3P) produces the 1-deoxy-D-xylulose 5-phosphate (DXP), catalyzed by the 1-deoxy-D-xylulose 5-phosphate synthase (DXS), the first committed step in the plastidial MEP-pathway. DXP is then reorganized and reduced to 2-C-methyl-D-erythritol 4-phosphate (MEP), by 1-deoxy-D-xylulose 5- phosphate reductoisomerase (DXR). MEP is then cyclized to the intermediate methylerythritol 2,4-cyclodiphosphate (ME-cPP), through the action of three consecutive enzymatic steps, involving a cytidylation step (by 4-Diphosphocytidyl-2-C-methyl-D-erythritol synthase, CMS), an ATP-dependent phosphorylation (by the CMK 4-(cytidine 5′-diphospho)-2-C-methyl-D-erythritol kinase), and a cyclization step (by the 2-C-methyl-D-erythritol 2,4-cyclodiphosphate synthase, MDS). The final two enzymatic steps of the MEP-pathway consist in the synthesis of the hydroxymethylbutenyl 4-diphosphate (HMBPP) from ME-cPP, catalyzed by the hydroxymethylbutenyl 4-diphosphate synthase (HDS) and the conversion of HMBPP into a 5:1 mixture of IPP and its isomer DMAPP, by the IPP and DMAPP synthase (IDS). According to the number of $C_5$ units of IPP and DMAPP, different isoprenoids are formed: in particular, diterpenes contain four $C_5$ units, characterized by a large diversity in their chemical structure, with more than 18,000 structures thus far described [5]. Basically, diterpenoids derive from three enzymatic steps: (i) condensation of four units of IPP and formation of the universal precursor GGPP (Geranyl geranyl diphosphate); (ii) the formation of very diversified chemical skeletons, catalyzed by different classes of diterpene synthases (diTPSs) [6], and (iii) the subsequent modification of the diterpene skeletons, catalyzed by cytochrome P450s (P450s or CYPs) [5], yielding a wide variety of complex chemical structures, including acyclic bi-tri- and tetra-cyclic compounds. The *Salvia* species are rich in tricyclic diterpenoids, and more than 400 diterpenoids with different abietane skeletons have been isolated from *Salvia* species [7].

Different biological properties for most plant diterpenoids have been reported, including antitumor [8,9], cytotoxic, antibacterial [10], antiplasmodial [11,12], leishmanicidal, gastroprotection, molluscicidal [13], and antifungal activities [14]. Some diterpenoids are currently used in the clinical practice, such as paclitaxel, ginkgolide, oridonin, tanshinones, and triptolide [15].

In this review, we present an integrated overview of the possible avenues for enhancing the biosynthesis of abietane diterpenes in *Salvia sclarea* hairy roots by either elicitation or metabolic engineering of the MEP pathway from which they derive.

**Figure 1. Simplified MEP-derived biosynthetic route of abietane diterpenes and other isoprenoids in Lamiaceae species**. The main enzymatic steps, intermediates, and final products are indicated. Abbreviations: MEP, 2-C-methyl-D-erythriol-4P; DXS, deoxyxylulose 5-phosphate synthase; DXR, deoxyxylulose 5-phosphate reductoisomerase; CMS, 4-diphosphocytidyl-methylerythritol synthase; CMK, 4-diphosphocytidyl-methylerythritol kinase; MCS, methylerythritol 2,4-cyclodiphosphate synthase; HDS, hydroxymethylbutenyl 4-diphosphate synthase; HDR, hydroxymethylbutenyl 4 diphosphate reductase; GPPS, GPP synthase; IDI, isopentenyl diphosphate isomerase; CPPS, CPP synthase; IPP, isopentenyl pyrophosphate; DMAPP, dimethylallyl pyrophosphate; GPP, geranyl pyrophosphate; GGPP, geranylgeranyl pyrophosphate; CPP, copalyl diphosphate; entCPP, ent-copalyldiphosphate; GAs, Gibberellins; ent*CPPS*, ent-copalyldiphosphate synthase; MiS, miltiradiene synthase; HFS, hydroxy-ferruginol synthase; $C_{20}$-Ox, $C_{20}$-oxidase. The terpene synthases CPPS and MiS cyclize GGPP to miltiradiene, which is converted spontaneously by oxidization to abietatriene. A HFS, a cytochrome P450 monooxygenases, oxidizes abietatriene to ferruginol and 11-hydroxyferruginol, through two subsequent oxidation steps. A further oxidation of 11-hydroxyferruginol position at the $C_{20}$, by a $C_{20}$-oxidase, produces carnosic acid. A possible biosynthetic route of aethiopinone, 1-oxo-aethiopinone and salvipisone is also shown. Dotted lines indicated unknown enzymatic steps. In red are indicated the biosynthetic genes whose expression has been modified by metabolic engineering in *S. sclarea* hairy roots.

## 2. Methodologies

This review is based on the main results published by our group on metabolic engineering of abietane diterpenes in *S. sclarea*, compared with publically available data on this specific topic in *S. sclarea* and other medicinal or crop plants. A bibliographic search was launched by using the keywords "*Salvia sclarea*", "elicitation", "metabolic engineering", "abietane diterpenes", "hairy roots", which yielded a total of 24,949 results, divided as follows: *Salvia sclarea*: 164 results; abietane diterpenes: 3152; hairy roots: 2093; plant metabolic engineering: 7931; the plant elicitation: 10,909 results. The search field was further restricted considering these same keywords coupled to "*Salvia sclarea*" keyword and considering also a search for the therapeutic relevance, the bioactivity, and the anti-tumor properties of abietane diterpenes. The use of these criteria permitted to have a collection of 1230 articles, published until June 2022. Non-relevant articles were excluded, and relevant studies were considered for writing the manuscript. The selection criteria of the collected articles are schematized in Figure 2.

**Figure 2.** Diagram representing bibliographic research criteria used for this review.

## 3. *S. sclarea* Roots Contain Bioactive Abietane Diterpenes

*Salvia sclarea* (clary sage) is a biennial or perennial plant belonging to the Lamiaceae family, typical of the north of the Mediterranean, central Asia, and some areas of North Africa. It is well known for the extraction of sclareol produced in the flower calyces, which is used for the semi-synthesis of the ambroxide, a well-prized perfume ingredient. This compound derives by the cyclization of GGPP by a Labd-13-en-8-ol diphosphate synthase (SsLPPS) to labda-13-en-8-ol diphosphate (LPP), which is subsequently converted into sclareol [16]. However, interesting additional abietane-quinone-type diterpenes are synthesized in the roots of *S. sclarea*, such as carnosic acid, aethiopinone, 1-oxoaethiopinone, salvipisone, and ferruginol, with known pharmacological properties, summarized in Table 1. Aethiopinone, salvipisone, 1-oxoaethiopinone, and ferruginol, from *S. sclarea*,

showed bacteriostatic as well as bactericidal activities against different strains of *Staphylococcus aureus* and *Staphylococcus epidermidis*, through a synergistic action of salvipisone and aethiopinone with ß-lactam antibiotics [17]. Cytotoxic and antitumor activities in several human tumor cell lines have been reported for tricyclic diterpenoids [9,18,19], especially those containing quinone moiety, often present in several effective cancer chemotherapeutic agents [20]. Salvipisone and aethiopinone showed relatively high cytotoxicity against HL-60 and NALM-6 leukemia cells, by inducing a caspase-3-mediated apoptosis [19]. In addition, our studies have contributed to establish that aethiopinone, purified from *S. sclarea* hairy roots, has also cytotoxic different and anti-proliferative activities against other tumor cells, in particular against solid tumor cell lines, such as MCF7 (breast adenocarcinoma), HeLa (epithelial carcinoma), PC3 (prostate adenocarcinoma), and A375 (human melanoma), for which drug resistance is often reported [21], and with negligible effects in non-tumor cells [9]. Especially interesting is this anti-melanoma activity exerted by aethiopinone, since melanoma cells are intrinsically resistant to pharmacological anticancer treatment and pharmaceutical companies are searching for more active and stable drugs [21].

One drawback in using these interesting compounds as potential new anti-tumor drugs is their very low amount (<0.5% dry weight) in natural and cultivated plants, as reported frequently for most of the plant secondary metabolites [1]. Plant cell and tissue cultures have been largely considered an attractive possibility of extraction of plant bioactive secondary metabolites [22]. In particular, hairy root (HR) technology has been reported to be an efficient system for producing secondary metabolites, especially for those which accumulate preferably in differentiated plant organs, such as roots [23]. A HR-based platform has been developed in our laboratory as a starting point to overcoming potential metabolic bottlenecks in the enzymatic reactions acting upstream or downstream of GGPP, and with the final aim to optimizing the biosynthesis of aethiopinone and other tricyclic abietane diterpenoids in *S. sclarea*, as summarized below.

**Table 1.** Abietane-quinone-type diterpenes and their reported biological activities in Lamiaceae species.

| Diterpene | Plant Species | Biological Activity | References |
|---|---|---|---|
| **aethiopinone** | *Salvia sclarea* *Salvia aethiopis* *Salvia lachnocalyx* | Antimelanoma Antimicrobial Cytotoxic Antiinflammatory | [9] [17] [19,24] [25] |
| **carnosic acid** | *Rosmarinus officinalis* *Salvia sclarea* | Neuroprotective Citotoxic Antioxidant | [26,27] [28–30] [31] |
| **ferruginol** | *Salvia amplexicaulis* *Salvia eriophora* *Salvia sclarea* | Cardioactive Neuroprotective Cytotoxic | [32–34] [35] [28,36,37] |
| **1-oxo-aethiopinone** | *Salvia sclarea* | Antimicrobial | [17,38] |
| **salvipisone** | *Salvia sclarea* | Antimicrobial Cytotoxic | [17,38] [19,28] |

It is worthy to note that several bioactive diterpenes produced in the roots of other *Salvia* species have been largely studied. For example, tanshinones, a class of lipophilic abietane diterpenes present in the roots of *S. miltiorrhyza*, known for their multiple therapeutic activities, have been deeply characterized, and their biosynthetic route from GGPP almost completely clarified [39].

Less is known on the enzymatic steps that from GGPP lead to the synthesis of abietane diterpenes synthesized in the roots of *S. sclarea*, which may hinder the possibility of genetically engineering genes encoding limiting enzymes with the final aim to increase the availability of substrates/intermediates/final products of this biosynthetic pathway. This is a quite relevant aspect not only for a cost-effective scale-up of the production of

this class of compounds, but even for a deeper understanding of their cellular targets and pharmacological characterization.

As reported in the following paragraphs, the establishment of stable HR lines of *S. sclarea*, combined with complementary strategies of metabolic engineering and elicitation, has contributed for the first time to establish that the biosynthesis of aethiopinone proceeds through the cyclization of GGPP to CPP mediated by the enzyme CPPS. We have also demonstrated that not only the availability of GGPP, as already reported for several different plant diterpenes, but also of CPP is quite critical to enhance the accumulation of aethiopinone and other abietane-type diterpenes in *S. sclarea* HRs [40].

## 4. A Hairy Root Platform for the Production of Bioactive Abietane Diterpenes

HRs are adventitious roots obtained at plant-wounded sites by the infection of *Rhizobium rhizogenes* (previously denominated *Agrobacterium rhizogenes*), a Gram-negative soil bacterium. HRs are differentiated organs, which grow under hormone-free culture conditions, and are considered a valuable and stable source of plant bioactive compounds more than plant cell cultures, which are frequently biochemically variable and unable to produce sufficient quantities of bioactive secondary metabolites, [23].

Stable massive cultures of *S. sclarea* HRs were obtained in our laboratory by customizing the growth media and conditions [9], in which the metabolic flux was modified toward a higher synthesis of aethiopinone and other abietane diterpenoids, as schematized in Figure 3.

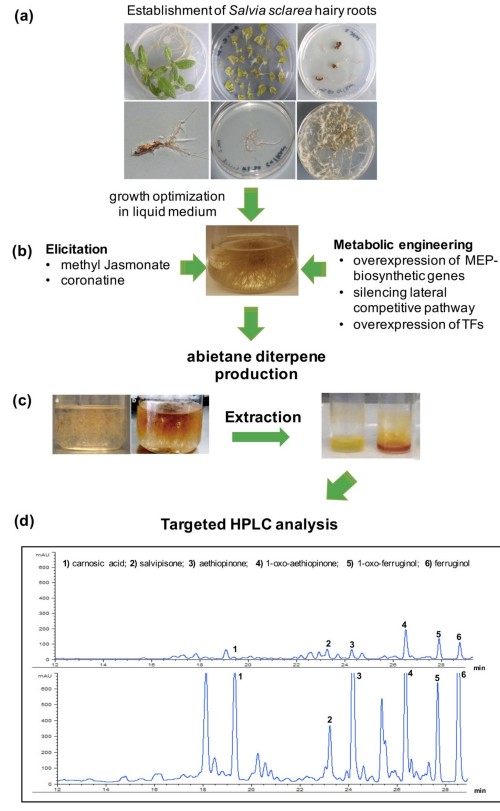

**Figure 3. Enhancing abietane diterpene content of *S. sclarea* by coupling hairy root technology and different elicitation or metabolic engineering approaches.** (**a**) Stable hairy root cultures obtained by transformation with *R. rhizogenes*; (**b**) elicitation with methyl jasmonate or coronatine and different metabolic engineering approaches used to increase the accumulation of abietane diterpenes, as phenotypically visible from the red color of the hairy roots and the methanol extracts; (**c**) representative HPLC-DAD chromatograms of the main abietane diterpenes identified in control *S. sclarea* HRs (upper panel) and HRs elicited with MeJA for 1 week (lower panel) (**d**). The photos and the chromatograms are original and used by the authors for this review.

## 5. Boosting the Biosynthesis of Abietane Diterpenes in *S. sclarea* HRs by Elicitation and Metabolic Engineering

In the last decade, our efforts were focused on trying to elucidate the metabolic pathway of abietane diterpenes in *S. sclarea* and to identify potential limiting enzymatic reactions and precursors/intermediates that might affect their accumulation in roots. This information was gathered by designing complementary strategies of elicitation and metabolic engineering to modify the expression of genes encoding known, unknown limiting or competitive enzymatic reactions of the MEP-pathway to boost the metabolic flux toward a higher synthesis of aethiopinone and other abietane diterpenoids, as recapitulated in Table 2.

**Table 2.** A summary of the effects on the abietane diterpene accumulation in *Salvia sclarea* HRs obtained by designing different elicitation and metabolic engineering approaches.

| Gene | Modification/ Elicitation | Total Abietane Diterpenes mg g$^{-1}$ Dry Weight | Fold- Increase | Aethiopinone Content mg g$^{-1}$ Dry Weight | References |
|---|---|---|---|---|---|
| HRs control | Not transformed/not elicited | $1.2 \pm 0.12$ | _ | $0.5 \pm 0.03$ | |
| *AtDXS* | overexpression | $\pm 0.22$ ** | 2.1 | $1.01 \pm 0.25$ * | [9] |
| *AtDXR* | overexpression | $3.38 \pm 1.02$ *** | 2.8 | $1.54 \pm 0.4$ ** | |
| *Ssent -CPPS* | RNA interference | $4.5 \pm 0.50$ *** | 3.7 | $1.86 \pm 0.11$ ** | |
| *SsGGPPS* | overexpression | $25.64 \pm 1.22$ *** | 21.3 | $7.8 \pm 0.71$ *** | [40] |
| *SsCPPS* | overexpression | $37 \pm 1.60$ *** | 30.8 | $10.44 \pm 0.21$ *** | |
| *SsGGPPS / SsCPPS* | overexpression | $10.24 \pm 0.68$ *** | 8.5 | $5.2 \pm 0.12$ *** | |
| cyanobacterial *DXS* | overexpression | $2.9 \pm 0.13$ ** | 2.4 | $1.21 \pm 0.11$ ** | [41] |
| cyanobacterial *DXR* | overexpression | $4.5 \pm 0.24$ *** | 3.75 | $2.27 \pm 0.22$ *** | |
| AtWRKY40 | overexpression | $4.8 \pm 0.43$ ** | 4.0 | $2.9 \pm 0.12$ ** | |
| AtWRKY18 | overexpression | $2.5 \pm 0.03$ * | 2.1 | $0.77 \pm 0.01$ * | [42] |
| AtMYC2 | overexpression | $6.5 \pm 1.62$ ** | 5.4 | $3.1 \pm 0.10$ *** | |

Mean values obtained from three independent adventitious hairy root lines for each elicitation treatment or transformation event (* $p \leq 0.05$, ** $p \leq 0.01$; *** $p \leq 0.001$). Data in the table are derived from the indicated publications of the authors.

### 5.1. Elicitation with Methyl Jasmonate and Its Analogue Coronatine

Elicitors are biotic or abiotic agents which trigger signal cascades inducing tolerance and immune response in plants. They have been widely applied to stimulate the synthesis of secondary metabolites in HRs as well as in plant cell culture of different plants [43,44]. Jasmonic acid (JA) and its structural analogue coronatine (Cor) have been extensively reported as elicitors of the MEP-pathway. Both JA or Cor, which structurally mimics JA-isoleucine (JA-Ile), the bioactive JA, bind the CORONATINE INSENSITIVE 1 (COI1), to activate JA signaling and the expression of JA-induced genes. Cor, however, is able to induce other defense pathways in an independent fashion from the JA signaling [45].

The COI1 dependent perception and signaling pathway is regulated by the transcription factor MYC2 that acts as activator and repressor of JA-responsive gene expression in *Arabidopsis* [46]. It is known that MYC2 is repressed by the JAZ proteins, which bind the co-repressor TOPLESS (TPL) protein, directly through the EAR (Ethylene Response Factor-Associated Amphifilic Repression) motif or indirectly through an additional repressor, the NINJA (Novel Interactor of JAZ) protein. The histone acetylases HDA6 and HDA19 are then recruited to form a closed complex inhibiting JA response (Figure 4a). Under a high level of JA-Ile (caused by stress conditions) or elicitation with Cor, JA-Ile or Cor are transported into the nucleus, where they interact with the F-box protein COI1 within the SCF E3 ubiquitin ligase (Skp–Cullin–F-box-type) and form the SCF-COI1 complex. This complex recruits the JAZ repressor proteins, bound to the promoter of MYC2, and sends them to degradation via the 26S proteasome. Released MYC2 binds the subunit 25 of Mediator complex (MED25) and induces the transcription of early JA-responsive genes [46],

as well as genes involved in the plant defense mechanisms, including also genes encoding enzymes of different secondary metabolites (Figure 4b) [47].

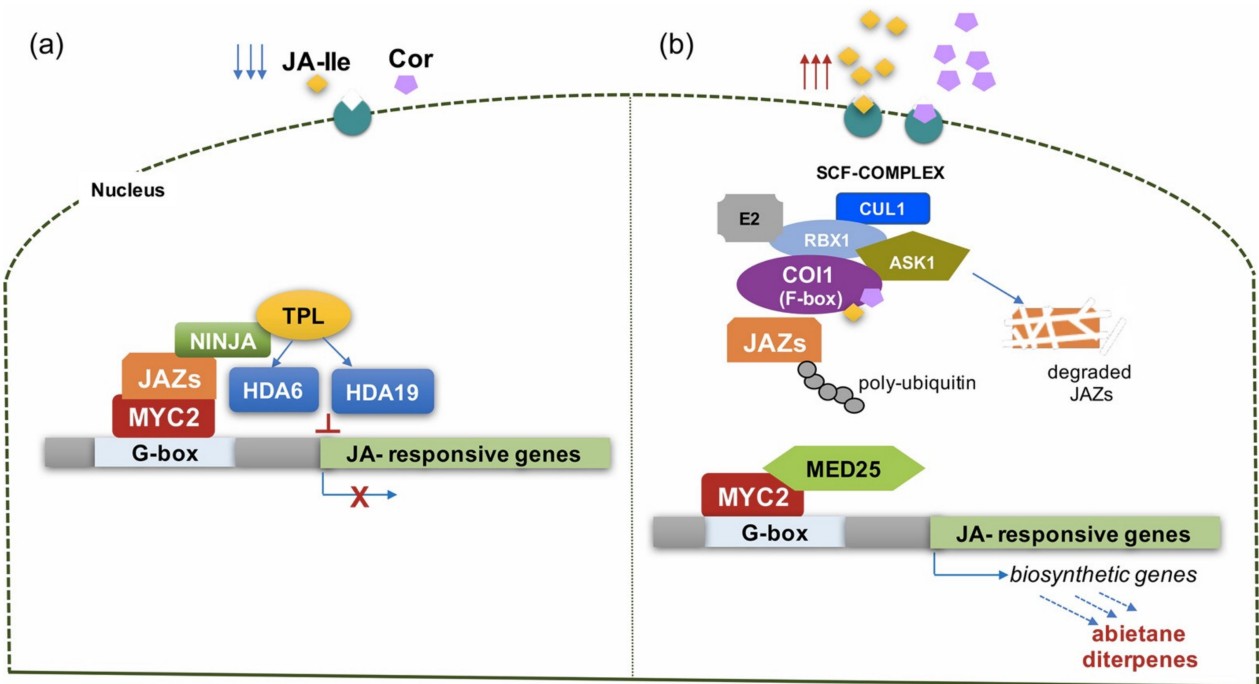

**Figure 4. A Schematic representation of the MJ or Cor signal perception and transduction mediated by the MYC2 transcription factor**, based on the current knowledge available in literature (original figure). A detailed description of MYC2-mediated signaling and transduction in normal conditions (low level of JA-Ile or Cor) (**a**) or in stress or elicitation conditions (high level of JA-Ile or Cor) (**b**) is reported in the text. Abbreviations: JA-Ile, jasmonoyl isoleucine; Cor, coronatine; JAZ, jasmonate ZIM domain; NINJA, novel interactor of JAZ; TPL, topless; HDA6, HDA19, histone deacetylase 6, 19; ASK1, Arabidopsis SKP1 (S-phase kinase-associated protein (1) homologue; CUL, CULLIN; E2, ubiquitin-conjugating enzyme; MYC2, bHLH zip transcription factor; RBX, RING-H2 protein; SCF-complex, complex consisting of Skp1, Cullin-1, and F-box protein; Ub, ubiquitin; COI1, coronatine insensitive 1; MED25, mediator 25.

Elicitation of *S. sclarea* HRs with Methyl-jasmonate (MJ) or Cor has been very informative for us to pinpoint biosynthetic genes of MEP-pathway that might limit the accumulation of aethiopinone and other bioactive diterpenoids in *S. sclarea* HRs, as reported below.

### 5.1.1. MJ-Induced Accumulation of Bioactive Abietane Diterpenes in *S. sclarea* Is Due to the Transcriptional Regulation of Genes of the MEP-Pathway

It is well documented that the JA-activated biosynthesis of different terpenoids [47], is due to transcriptional up-regulation of genes belonging to either the MEP-pathway or the MVA-pathway [48–50].

We found that also in *S. sclarea* HRs, MJ elicitation was able to enhance the transcriptional levels of several biosynthetic genes of the plastidial MEP-dependent pathway, such as *SsDXS*, *SsDXR*, *SsCMK*, *SsMCS*, *SsHDS*, *SsHPR*, *SsGGPPS*, and *SsCPPS* genes [51]. The most induced transcript by MJ was *SsDXS* (60-fold increase), the enzyme acting up-stream the MEP-derived pathway. Since it is well known that this enzymatic reaction is limiting the accumulation of different isoprenoids in many plant species, increasing the DXS activity by metabolic engineering has been successful to enhance the content of terpenoids in different plant species and prokaryotic cells [52]. The *SsGGPPS* gene encoding the synthase involved in the condensation of IPP and DMAPP to GGPP, the common precursor of many terpenes of plastidial origin, was also transcriptionally activated by MJ (20-fold increase) in *S. sclarea* HRs as well as the *SsCPPS* gene (60-fold increase). This is indirect

evidence that abietane-type diterpenes in *S. sclarea* are produced from the conversion of GGPP via a copalyl-diphosphate (CPP) intermediate, by using this CPP synthase, as widely reported for other plant diterpenoids [53–55]. This MJ-coordinated expression of multiple genes of the MEP-pathway induced a significant increase of the abietane diterpene content in *S. sclarea* HRs, as phenotypically visible from the red color of the MJ-elicited HRs (Figure 3). Compared to control HRs (14.50 ± 0.31 mg L$^{-1}$), MJ elicitation (100 μM) for 7 days induced an approximatively 20-fold enhancement in the total abietane diterpene content (304.01 ± 2.21 mg L$^{-1}$). A long-term elicitation (4 weeks) with MJ also increased the content of total abietane diterpenoids, but it was associated with a severe HR growth inhibition, which penalized the final yield of total abietane diterpenes (210.74 ± 1.26 mg L$^{-1}$) (Figure 5) [51].

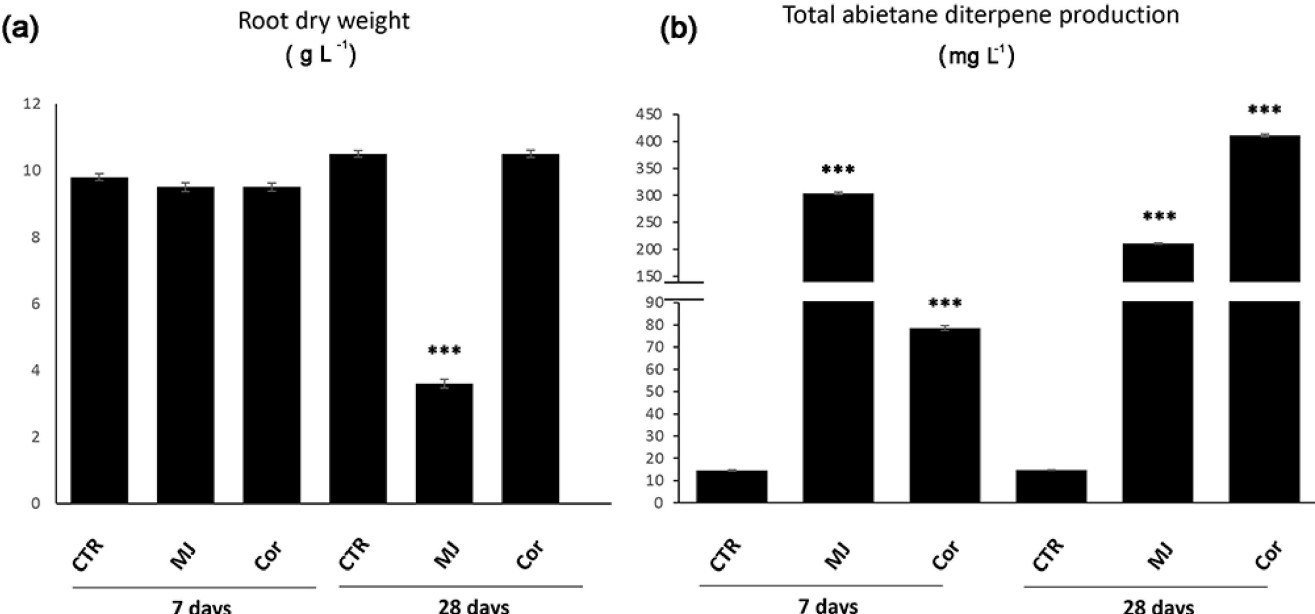

**Figure 5.** Effects of MJ or Cor elicitation (7 or 28 days) on the final biomass (**a**) and on the total abietane diterpene production in *S. sclarea* hairy roots (**b**). The asterisks denote a significant difference between elicited and control hairy roots ($p < 0.001$) according to Student's t test. Data are from Vaccaro et al. 2017.

This set of data are consistent with those reported in *Salvia milthiorriza* HRs elicited with MJ, which triggered the accumulation of tanshinone and phenolic acids [56], due also to the up-regulation of biosynthetic genes [57]. MJ has been also proved to induce simultaneously the transcription of most of the biosynthetic genes in specific pathways of other plant species, as reported for the TIA (terpenoid-indole-alkaloid) in *C. roseus* [58], for nicotine in *N. tabacum* [59], and for artemisinin in *A. annua* [60]. This MJ-concerted transcriptional activation of many genes encoding enzymes of the MEP-pathway has prompted numerous studies for identifying "master" regulators of this pathway. Different families of plant transcription factors have been identified as able to regulate the biosynthesis of terpenoids [61], as discussed in the Section 5.2.3.

### 5.1.2. Coronatine as Alternative Elicitor of the Synthesis of Abietane Diterpenes

Coronatine (Cor) is a structural analogue of the bioactive conjugate of the JA-leu, the active intracellular form of JA, capable of mediating various aspects of bacterial virulence. It causes the stomata to reopen to facilitate bacterial invasion, their growth in the apoplast, and the induction of disease symptoms [62]. There are several features that make Cor preferable to MJ as elicitor agents in plant cell/HR cultures: it is a more stable molecule, and does not need to be chemically converted to JA-Ile to bind the COI1 receptor [63]. *S. sclarea* HRs

elicited continuously for 7 or 28 days with Cor (0.1 µM) showed a significant transcriptional activation of different genes encoding biosynthetic enzymes of the MEP-pathway, although to a lesser extent compared to the MJ treatment. However, similarly to MJ treatment, the transcriptional gene activation was transient, peaking at 24 h from the beginning of the Cor elicitation [51]. Consistent with this coordinated transcription of biosynthetic genes, the total amount of abietane diterpenes was significantly enhanced in Cor-elicited *S. sclarea* HRs. After 7 days compared to the untreated HRs (14.50 ± 0.31 mg L$^{-1}$), a significant increase was observed in total abietane diterpene production (78.56 ± 1.12 mg L$^{-1}$, approximatively a 5-fold increase). After 28 days of Cor elicitation, the content of aethiopinone in *S. sclarea* HRs increased further (103.32 ± 2.10 mg L$^{-1}$, approximatively a 24-fold increase over the basal content of control hairy roots (4.40 ± 0.13 mg L$^{-1}$). Very low concentrations (0.1 µM) of Cor for 28 days allowed to extract 410.97 ± 2.50 mg L$^{-1}$ of abietane diterpenes, determining a 28-fold increase compared to the content of unelicited HRs, significantly higher than the increase observed after a long-term elicitation with MJ (Figure 5b). This was due mainly to a negligible negative effect caused by Cor treatment on the final HR biomass (Figure 5a) [51].

*5.2. Metabolic Engineering to Enrich the Content of Abietane Diterpenes in S. sclarea Hairy Roots*

Plant metabolic engineering is a powerful strategy to modify metabolic routes aimed at increasing the synthesis of a specific bioactive compound, by either genetic engineering of endogenous genes, with the final aim to increase or divert the metabolic flux towards desired/undesired products, respectively. Another application is to generate plants able to synthesize novel compounds through transformation with genes from other plants/organisms [64].

Our previous elicitation studies evidenced a strong correlation between the accumulation of abietane diterpenes and the level of expression of several biosynthetic genes of the MEP-pathway [51]. In the next section, we summarize the different approaches we have used and the results achieved in boosting the synthesis of abietane diterpenes in *S. sclarea* HRs by: i) tuning the expression of biosynthetic genes that control rate-limiting enzymatic steps (*DXS, DXR, GGPPS, CPPS* alone or in combination); ii) RNAi silencing of a gene acting at the lateral competitive route of gibberellin (*Ent-CPPS*); iii) overexpressing different transcription factors (*WRKYs* and *MYC2*) involved in the coordinated regulation of several genes of the MEP-pathway.

5.2.1. Overexpression of *DXS* and *DXR* Genes, encoding the First Two Enzymes Acting Up-Stream the MEP-Pathway

As reported previously, Deoxyxylulose 5-phosphate synthase (DXS), the first enzymatic step of the MEP-pathway, has been often reported as one of the limiting enzymes of the up-stream pathway to GGPP, by influencing the supply of IPP and its isomer DMAPP (extensively and recently reviewed by [65]). Experimental evidence has indicated that also the activity of DXP reducto-isomerase (DXR), the second enzymatic step, is limiting the biosynthesis of several plant isoprenoids [66]. The expression of the *DXS* and *DXR* genes has been modified in different plant species to enhance successfully the synthesis of different isoprenoids, although the reported results are not always consistent in different plant species [67–69]. In our earlier studies, we have demonstrated that constitutive overexpression of the heterologous *A. thaliana AtDXS* or *AtDXR* genes is able to enhance the content of bioactive abietane-type diterpenoids in *S. sclarea* HRs [9]. As reported in Table 2, overexpression of the DXS protein triggered a 2-fold increase in aethiopinone content compared to the control HR lines. However, high levels of the exogenous DXS protein caused a negative pleotropic effect on the HR growth, probably due to a general competition with the basal protein synthesis or to the modification of the content of other MEP-derived phytohormones (cytokines, gibberellin or ABA), which might interfere with normal HR growth and development. Interestingly, the overexpression of the *AtDXR* gene appeared to be slightly more efficient in enhancing the synthesis of abietane-type

diterpenes in *S. sclarea* HRs (a 3-fold increase compared to the control HR line), coupled to no detrimental effects on HR growth [9]. Overexpressing both genes in *S. miltiorrhiza* HRs has improved the accumulation of tanshinones by [68] as well as of other terpenoids in different plant species [69].

In a parallel study, we also tested whether overexpression of *DXS* and *DXR* genes of cyanobacterial origin might offer some biotechnological advantages in increasing the synthesis of bioactive diterpenes in *S. sclarea* HRs. Indeed, the introduction in a plant genome of heterologous genes by genetic transformation might avoid potential gene silencing or co-suppression events, often reported when overexpressing homologous genes, and, at the same time, ensuring a possible greater stability and activity of the two enzymes in a plastidial environment. Orthologous genes, amplified by the genomic DNA of *Synechocystis* sp PCC6803, were overexpressed in *S. sclarea* HRs, by targeting them to the chloroplast by the rbc plastid transit peptide [41]. An increased accumulation of bioactive diterpenes was obtained, and, interestingly, the overexpression of the bacterial *DXR* gene triggered a 5-fold accumulation of aethiopinone, significantly greater than the increase obtained by overexpressing the plant *AtDXR* gene (a 3-fold increase).

Altogether, these data corroborate the consolidated notion that DXR and DXS are limiting enzymatic steps of the MEP-pathway in several plant species, possibly directing the metabolic flux toward a higher availability of IPP and DMAPP. However, these two enzymes only partially explain the robust increase in abietane diterpenes we found in elicited HRs by MJ and Cor. Therefore, we focused our further studies on establishing other potential enzymatic bottlenecks in the MEP-pathway that might limit their accumulation in *S. sclarea* HRs.

### 5.2.2. GGPP Availability Limits the Accumulation of Abietane Diterpenes in *S. sclarea* HRs

GGPP is the universal isoprenoid precursor and it has been postulated that modifying its distribution among the different downstream metabolic branches of the plastidial MEP-pathway might direct the metabolic flux towards a higher biosynthesis of a targeted end product [70]. We proved that this is also true for an enhanced abietane diterpene accumulation in *S. sclarea* HRs, by chemically inhibiting the competitive gibberellin (GA) route with the 2-chloroethyl-N,N,N-trimethyl-ammonium chloride (CCC), a known inhibitor of entCPPS, the first enzymatic step, that from GGPP leads to gibberellins (GA) biosynthesis, or by RNAi-mediated silencing of the *ent-CPPS* gene (Figure 3). The block of this competitive metabolic pathway enhanced the content of all analyzed diterpenes (carnosic acid, ferruginol, 1-oxo-ferruginol, salvipisone, aethiopinone, and 1-oxo-aethiopinone) in CCC-treated HRs compared to the control HRs, with the most relevant increase in the content of aethiopinone (a 7-fold increase). RNAi-mediated silencing of the *entCPPS* gene confirmed that blocking this lateral GA route from GGPP is an efficient strategy to increase the abietane diterpene content in *S. sclarea* HRs [40] (Table 2). These data indirectly demonstrated that the GGPP pool is limiting in the metabolic route to abietane diterpenes, as already suggested by the significant correlation between the expression level of the *GGPPS* gene and the abietane diterpene content we have found in *S. sclarea*-elicited HRs [51]. These findings also pointed to the GGPPS enzyme as a potential target for increasing the biosynthesis of abietane diterpenes in *S. sclarea* HRs. By assuming that the IPP and DMAPP, the immediate GGPP precursors, would not be limiting, the cDNA full-length *SsGGPPS* gene, including its own plastid transit peptide, was constitutively overexpressed in *S. sclarea* HRs. Higher transcript levels of the *SsGGPPS* gene significantly increased the content of aethiopinone ($7.80 \pm 0.71$ mg g$^{-1}$ dw), salvipisone ($5.57 \pm 0.78$ mg g$^{-1}$ dw), and ferruginol ($8.51 \pm 0.35$ mg g$^{-1}$ dw), which was approximately 8, 7, and 28 times, respectively, higher than that in the control HR line ($0.99 \pm 0.08$ mg g$^{-1}$ dw). Altogether, these results indicate indirectly that the GGPP pool in *S. sclarea* HRs might limit the biosynthesis of abietane diterpenes, and that GGPPS overexpression is an efficient strategy to increase the overall content of this class of compounds in *S. sclarea* HRs [40].

### 5.2.3. CPP Is a Precursor of Abietane Diterpenes in *S. sclarea* and Overexpression of *SsCPPS* Also Enhances Their Content

The biosynthetic pathway of the abietane diterpenes synthesized in *S. sclarea* roots is poorly understood, although their chemical structures were established more than two decades ago [38,71]. For instance, little is known on the first GGPP cyclization step operated by diterpene synthases (diTPSs) in *S. sclarea* roots to yield abietane diterpenes. It is well documented that in conifers/gymnosperms, the formation of diterpenes requires an initial double cyclization of GGPP into (+)-copalyl diphosphate [(+)-CPP], operated by the class II active site of the abietadiene synthase, a bifunctional diTPS. The (+)-CPP intermediate is then modified by the class I diTPS active site and is subjected to a second cyclization enzymatic step, followed by structural rearrangements via intermediate carbocations [54]. In Lamiaceae species, it has been reported that the synthesis of labdane-related diterpenes also proceeds through the cyclization of GGPP to CPP, catalyzed by the copalyl diphosphate synthase (CPPS), a class II diTPS [72]. The following enzymatic reaction in the biosynthesis of this group of diterpenes is due to the action of a kaurene synthase-like enzyme, the miltiradiene synthase (MiS), which, through a spontaneous oxidation reaction, transforms CPP to miltiradiene.

A first indirect indication of the potential involvement of the CPPS in the cyclization of GGPP to CPP in the biosynthesis of *S. sclarea* abietane diterpenes was provided by the high correlation we found between the expression level of the *CPPS* gene and the abietane diterpene content in *S. sclarea* HRs [51]. These findings were further confirmed by the *SsCPPS* overexpression, which boosted a significant 20-fold increase in the aethiopinone content ($10.44 \pm 0.21$ mg g$^{-1}$ dw) compared to the control HR lines ($0.5 \pm 0.03$ mg g$^{-1}$ dw). Interestingly, the content of ferruginol ($11.50 \pm 0.38$ mg g$^{-1}$ dw) and salvipisone ($6.65 \pm 0.72$ mg g$^{-1}$ dw) was also enhanced by 30 or 9 times, respectively, by overexpressing the *CPPS* gene in *S. sclarea* HRs [40].

To the best of our knowledge, this is the first evidence that aethiopinone in *S. sclarea* might be synthesized from GGPP to CPP by a CPPS, which occurs for other abietane diterpenes [73,74]. In addition, the availability of the CPP might limit the synthesis of abietane diterpenes in *S. sclarea*, since we have proved for the first time that the *CPPS* overexpression overcomes this metabolic bottleneck.

### 5.2.4. Co-Expression of *GGPPS* and *CPPS* Genes

Despite the success obtained by overexpressing single genes, frequently, the final amount of a targeted metabolite is limited by multiple enzymatic activities. Therefore, the concerted regulation of two or multiple genes would be more appropriate to significantly obtain higher content of a desired compound. Co-expression of rate-limiting biosynthetic genes has been successfully applied to boost the synthesis of a variety of high-value plant-derived compounds [69,75]. Considering the significant increase in aethiopinone content obtained by overexpression of *GGPPS* and *CPPS* genes individually, these two biosynthetic genes were co-expressed in *S. sclarea* HRs [40]. This "push and pull" strategy induced an increase in the aethiopinone content by 6 times, against the 8- and 10-fold increase obtained by overexpressing the two genes singularly (Table 2). This might be probably caused by an unbalanced simultaneous high level of *GGPPS* and *CPPS* transcripts, and an impaired ratio of the levels of the two encoding enzymes and/or relative substrates. To better understand these unexpected results, it would be useful to determine the relative level of GGPPS and CPPS proteins and their relative enzymatic activities in order to identify overexpressing HR lines with a balanced level of both enzymes, which could affect additively or synergistically the final yield of aethiopinone and other bioactive diterpenes in *S. sclarea* HRs [40].

### 5.2.5. Orchestrating the Expression of Multiple Biosynthetic Genes of the MEP-Pathway by TFs

Transcription factors (TFs) provide an attractive alternative for modifying a metabolic flux in plants since they are able to activate in a coordinated manner the transcription of multiple biosynthetic pathway genes. It is widely reported that TFs may act alone

or in combination with other regulators to activate and/or to inhibit/de-repress gene transcription [76]. A bioinformatic analysis of *A. thaliana* genes encoding enzymes of the MEP-pathway revealed that the promoter of many of these genes contains the W-box (*TTGAC*), the known binding site of WRKY TFs, and the G-box (*CACGT*), the binding domain of MYC2, a TF known to be regulated by MJ (Figure 4). The WRKY TFs, members of a plant-specific TF family, characterized by a conserved peptide motif WRKYGQK and a zinc finger domain, play different biological roles and have a prominent involvement in controlling the plant response to biotic and abiotic constraints [77,78]. Among the different WRKY TFs thus far identified, we focused our attention on AtWRK18 and AtWRK40, since we identified in their promoter region multiple conserved methyl jasmonate responsive elements (MJRE), localized at −800 bp from the TSS, and confirmed their transcriptional activation in MJ-elicited Arabidopsis plantlets [42]. The overexpression of At*WRKY18*, At*WRKY40* genes in *S. sclarea* HRs positively activated the transcription of different biosynthetic genes of the plastidial MEP-derived pathway. In particular, *S. sclarea* genes *DXS*, *DXR, HDS, GPPS*, and *CPPS* were up-regulated by overexpression of *AtWRKY18* and *AtMYC2*, while At*WRKY40* preferentially activated the transcription of the *DXS* and *CPPS* genes. Targeted metabolic profiling of overexpressing WRKY HR lines revealed that the simultaneous up-regulation of the biosynthetic genes correlated to an increased content of abietane diterpenes compared to control line (a fold-increase >4 and >2 in the At*WRKY40* and *AtWRK18* overexpressing lines, respectively), especially in the production of salvipisone and aethiopinone [42]. This is one of the first published evidence on the involvement of WRKY TFs in the specific metabolic pathway of abietane diterpenes in *S. sclarea*, which was confirmed also for the biosynthesis of tanshinones in *S. miltiorrhyza* [79].

We also overexpressed in *S. sclarea* HRs the AtMYC2 TF, which contains a basic helix-loop-helix (bHLH) domain, and, as already discussed, is a master regulator of the JA signaling pathway and also controls secondary metabolism in plants [80–82] (Figure 4). *AtMYC2* overexpression was able to enhance the transcript levels of several biosynthetic genes of the MEP-derived pathway [42]. This coordinated up-regulation resulted in a significant accumulation of the total abietane diterpene content (a fold-increase > 5) (Table 2), which caused however, a considerable growth inhibition of the overexpressing HRs [42], due possibly to the multiple roles of MYC2 in JA signal transduction. As mentioned earlier, a possible unexpected modification of the level of the synthesis of phytohormones or chlorophyll derived from the MEP-pathway might also explain partially the growth impairment of the MYC2 overexpressing HR lines. At any rate, the MYC2- or WRKY-dependent increased content of abietane diterpenes was lower than that obtained by MJ or Cor elicitation, suggesting that it is more likely a combinatorial role for different TFs, belonging to the same TF group or to different families, in the transcriptional regulation of biosynthetic genes of JA-mediated metabolic pathways [42].

## 6. Conclusions and Perspectives

Our results demonstrate that the combined application of massive HR culture and metabolic engineering and/or elicitation strategies can contribute to bypass metabolic constraints in the MEP-pathway and improve the accumulation of bioactive abietane diterpenes in *S. sclarea*, and they might be successful applied as well as in other *Salvia* species. We are, however, still very far from a full comprehension of the molecular players involved in the regulation of the biosynthesis of abietane diterpenes in the roots of *S. sclarea*. Enzymatic steps downstream CPP to aethiopinone have not been elucidated yet and we are only at the beginning of understanding the complex TF network controlling positively or negatively this pathway.

In general terms, it is now evident that the regulation of plant secondary metabolism is not simply an on/off switch of single or multiple genes. New emerging technologies and the disclosure of novel molecular mechanisms regulating the plant secondary metabolism will give a great impulse to basic and translational research in this field, such as:

i. the possibility of engineering enzymes, removing catalytic constraints in key enzymes or knocking repressors, by CRISPRS-Cas9 genome editing (as reviewed in [83];

ii. many secondary metabolites are synthesized/accumulated in response to a plethora of abiotic and biotic stress. Epigenetic control of plant stress response is well documented, but the role of epigenetics in regulating the secondary metabolism has so far been largely overlooked [84].

It is expected that in the near future, the novel knowledge globally gathered on the regulation of plant secondary metabolism will lead to biotechnological innovations for the production of molecules of plant origin with high added value, from massive cell and tissue cultures or by reconstructing metabolic pathways in heterologous systems, according to the principles of sustainability, economy, and standardization required by the pharmaceutical industry and the market.

**Author Contributions:** A.L. and M.A. conceived and structured this review paper, based on experimental results designed and carried out by A.L., M.A., M.V., A.A. and N.D.T. All authors have read and agreed to the published version of the manuscript.

**Funding:** This research received no external funding.

**Conflicts of Interest:** The authors declare no conflict of interest.

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
