# Peer review of "Overcoming Metabolic Constraints in the MEP-Pathway Enrich Salvia sclarea Hairy Roots in Therapeutic Abietane Diterpenes"

_applsci, doi:10.3390/app12147116_

Round 1

Reviewer 1 Report

In the manuscript entitled “Overcoming metabolic bottlenecks in the MEP-pathway to enrich Salvia sclarea hairy roots in therapeutic abietane diterpenes” the authors provide a comprehensive review, primarily focused on their published work in optimizing the biosynthesis of abietane diterpenoids in the hairy roots of Salvia sclarea. While this plant has proven significant medicinal value, the extracts that can be obtained from the roots in vivo are limited. The authors discussed their work in which they obtained stable Salvia sclarea hairy roots through methyl jasmonate or coronatine elicitation. They further discussed how they overexpressed genes including DXS, DXR, GGPPS, CPPS that regulate the rate limiting steps in the MEP pathway, or transcription factors such as AtWRKY40, AtWRKY18 and AtMYC2 and silence the lateral competitive gene, (Ent-CPPS) to increase abietane diterpene content in Salvia sclarea hairy roots. Through this combination of approaches, they successfully enhanced abietane diterpene content from 2 to 30 fold compared to wild-type levels.

This manuscript is well-written, clearly explained and easy to understand. I did not find any significant issues with the review. I only have one suggestion for figure 2. The authors should include additional labels on this figure, particularly the HPLC-DAD chromatogram, it is not clear just by looking at the figure what it is showing.

Reviewer 2 Report

The manuscript under the title " Overcoming metabolic bottlenecks in the MEP-pathway to enrich Salvia sclarea hairy roots in therapeutic abietane diterpenes" can be accepted after revising the following comments:

- The title is unclear not relevant to the content. what is meant by "bottlenecks" exactly?

- Figure 2 (d) legend should be corrected.

- What is the source of figure (2)?

- the conclusion is very long need to be summarised 

Reviewer 3 Report

Mderate edit need
